# On Convergence of Nearest Neighbor Classifiers over Feature Transformations

**Luka Rimanic**[*]
ETH Zurich
luka.rimanic@inf.ethz.ch

**Cedric Renggli**[*]
ETH Zurich
cedric.renggli@inf.ethz.ch

**Bo Li**
UIUC
lbo@illinois.edu

**Ce Zhang**
ETH Zurich
ce.zhang@inf.ethz.ch

## Abstract

The k-Nearest Neighbors (kNN) classifier is a fundamental non-parametric machine learning algorithm. However, it is well known that it suffers from the *curse of dimensionality*, which is why in practice one often applies a kNN classifier on top of a (pre-trained) feature transformation. From a theoretical perspective, most, if not all theoretical results aimed at understanding the kNN classifier are derived for the *raw* feature space. This leads to an emerging gap between our theoretical understanding of kNN and its practical applications.

In this paper, we take a first step towards bridging this gap. We provide a novel analysis on the convergence rates of a kNN classifier over transformed features. This analysis requires in-depth understanding of the properties that connect *both* the transformed space and the raw feature space. More precisely, we build our convergence bound upon two key properties of the transformed space: (1) *safety* – how well can one recover the raw posterior from the transformed space, and (2) *smoothness* – how complex this recovery function is. Based on our result, we are able to explain why some (pre-trained) feature transformations are better suited for a kNN classifier than others. We empirically validate that both properties have an impact on the kNN convergence on 30 feature transformations with 6 benchmark datasets spanning from the vision to the text domain.

## 1 Introduction

The k-Nearest Neighbor (kNN) algorithms form a simple and intuitive class of non-parametric methods in pattern recognition. A kNN classifier assigns a label to an unseen point based on its $k$ closest neighbors from the training set using the maximal vote [1]. Even its simplest form, the 1NN classifier, converges to an error rate that is at most twice the *Bayes error* – the *minimal* error of any classifier [10]. Furthermore, when $k = k_n$ is a sequence satisfying $k_n/n \to 0$, as $n \to \infty$, the kNN classifier is consistent, meaning that its error converges to the Bayes error almost surely [34]. In recent times, kNN algorithms are popular, most often due to their simplicity and valuable properties that go beyond accuracy: (a) evaluation of Shapley value in polynomial time, used to outperform a range of other data valuation algorithms, whilst being orders of magnitude faster [19, 20, 21]; (b) estimation of the Bayes error [10, 12, 33]; (c) robustness analysis [28, 39]; (d) efficiency in enabling tools such as provable robust defenses [41]; (e) polynomial-time evaluation of the exact expectation of kNN over a tuple-independent database [24], which is generally hard for other classifiers; (f) applications to conformal inference tasks benefit from the fact that no training is required for running kNN [30].

---

[*]Equal contribution.

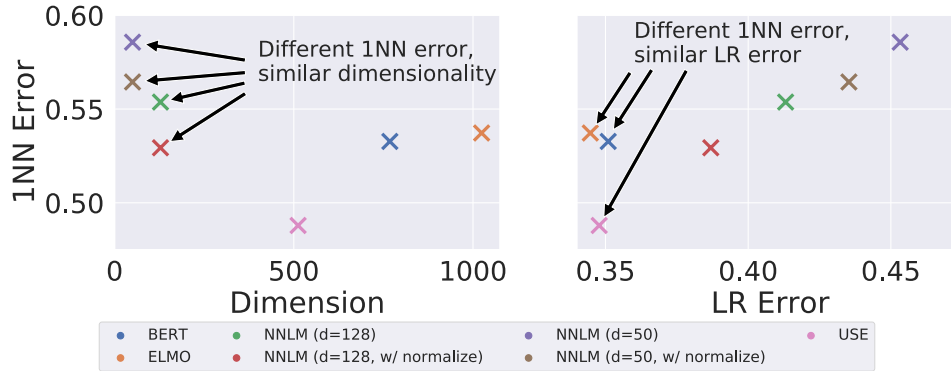

Figure 1: Challenges of examining the behavior of the 1NN classifier on top of feature transformations on YELP dataset. **(Left)** 1NN vs. dimension, **(Right)** 1NN vs. LR Error.

However, being itself a simple classifier, most of the above applications require kNN to be run on top of a feature transformation, ranging from simpler ones, such as PCA, to more complex transformations, such as neural networks pre-trained on another task.

At the same time, most, if not all, theoretical results on kNN are derived under assumptions that the algorithms are directly applied on the raw data, resulting in an *emerging gap between the theoretical understanding of kNN and its practical applications.* In this paper we take a step towards closing this gap. Specifically, we are interested in the *convergence behavior* of kNN over transformations: *Given a transformation $f$ and a set of $n$ training examples, can we describe the (expected) error of kNN over $f$ as a function of $n$ and some properties of the transformation $f$?* In other words, given a fixed $n$, what are the properties of $f$ that strongly correlate with the kNN error over $f$?

**Challenges.** Giving an informative answer to these questions is nontrivial as many simple, intuitive hypothesis alone cannot fully explain the empirical behavior of kNN over feature transformations.

First, as many existing results on the convergence have a factor of $(k/n)^{1/D}$, where $D$ is the dimension of the space, a natural hypothesis could be: *given a fixed $n$, transformations resulting in lower $D$ have lower kNN error.* While this is plausible, it leaves many empirical phenomenons unexplained. For example, Figure 1 (left) illustrates kNN errors over feature transformations with different dimensions, showing a real-world dataset in which transformations of the same dimension have drastically different kNN accuracies. Another approach is to consider the *accuracy* of some other classifier. For example, training a logistic regression (LR) model on $f$ and using it directly to form a hypothesis: *given a fixed $n$, transformations that lead to higher logistic regression accuracy have lower kNN error.* This is a much stronger hypothesis and can explain many scenarios that dimensionality alone cannot. However, Figure 1 (right) shows that this still leaves some important cases unexplained, providing examples in which multiple transformations achieve similar logistic regression accuracies, whilst having noticeably different kNN accuracies.

**Summary of Results and Contributions.** In this paper we take a step towards understanding the behavior of a kNN classifier over feature transformations. As one of the first papers in this direction, our results by no means provide the *full* understanding of *all* empirical observations. However, we provide a novel theoretical understanding that explains more examples than the above notions and hope that it can inspire future research in this direction. Our key insight is that the behavior of kNN over a feature transformation $f$ relies on two key factors: (1) *safety – how well can we recover the posterior in the original space from the feature space*, and (2) *smoothness – how hard it is to recover the posterior in the original space from the feature space?*

We answer these two questions by defining and examining *safety* as the decrease in the best possible accuracy of any classifier (i.e., the Bayes error) in the feature space when compared to the original features, whilst we use the geometrical nature of the transformed features to examine *smoothness*.

More precisely, let $\mathcal{X}$ be the feature space and $\mathcal{Y}$ the label space. In line with previous work on convergence rates of a kNN classifier, throughout the theoretical analysis we restrict ourselves to binary classification with $\mathcal{Y} = \{0, 1\}$. For random variables $X, Y$ that take values in $\mathcal{X}, \mathcal{Y}$ and are jointly distributed by $p(x, y) = p(X = x, Y = y)$, let $\eta(X) = p(1|X)$ be the true posterior probability.

The main task is to recover $\eta(X)$ by $(g \circ f)(X)$, where $g$ is a trainable function for a fixed architecture, and $f$ is a feature transformation. We show that the above notions can be accordingly bounded by a function involving the $L^2$-error defined by $\mathcal{L}_{g,X}(f) = \mathbb{E}_X((g \circ f)(X) - \eta(X))^2$, and $L_g$, a Lipschitz constant of $g$. In particular, we prove that the convergence rate of a kNN classifier over a feature transformation $f$ is upper bounded by $\mathcal{O}(1/\sqrt{k} + L_g \sqrt[d]{k/n} + \sqrt[4]{\mathcal{L}_{g,X}(f)})$. The result depicts a trade-off between the safety of a feature transformation, represented by $\mathcal{L}_{g,X}(f)$, and the smoothness, represented by $L_g$. For example, the most common implementation of transfer learning is given by $g(x) = \sigma(\langle w, x \rangle)$, where $\sigma$ is the sigmoid function, in which case one can take $L_g = ||w||_2$. We show that with this formulation we can explain the relative performance of many transformations used in the kNN setting. An important insight one might take is the following: *For two transformations that have similar $\mathcal{L}_{g,X}(f)$, the one with smaller $||w||_2$ will likely achieve better performance with respect to a kNN classifier.*

We highlight the usefulness and validate our novel theoretical understanding by conducting a thorough experimental evaluation ranging over 6 real-world datasets from two popular machine learning modalities, and 30 different feature transformations.

## 2 Related Work

As one of the fundamental machine learning models, the kNN classifier enjoy a long history of theoretical understanding and analysis. The first convergence rates of a kNN classifier were established in [9], where convergence rates of 1NN in $\mathbb{R}$ under the assumption of uniformly bounded third derivatives of conditional class probabilities were given, further extended to $\mathbb{R}^d$ in [14, 38]. Distribution-dependant rates of convergence have been examined in [17, 26], where certain smoothness assumptions were imposed. These were further developed in [7] through the notion of effective boundary, similar to the margin and strong density conditions examined in [3, 13, 15, 25, 37]. Finally, [18] provides convergence rates under the simplest assumptions, which perfectly suit our purposes. To the best of our knowledge, all previous works on the rates of convergence of kNN classifiers are derived on raw features. The goal of this work is to understand the convergence of kNN classifiers on transformed features, an emerging paradigm that we believe is, and will be, of great importance in practice.

It is no secret that kNN classifiers over raw features are cursed, suffering from what is known as the *curse of dimensionality* [10, 32, 33]. In practice, there have been many results that apply kNN classifiers over feature transformations. In the easiest form, such transformations can be as simple as a PCA transformation [23, 27, 36]. Recently, a popular choice is to apply kNN over pre-trained deep neural networks [4, 19, 28, 40]. Other related works propose to optimize a neural feature extractor explicitly for a kNN classifier [16, 42]. Most of these results show significant improvements on accuracy, empirically, bringing the performance of a simple kNN classifier on a par with state-of-the-art models. However, there is an obvious lack of rigorous understanding of properties that a feature transformation needs to satisfy in order to achieve a good kNN performance. This work is inspired by the empirical success of applying kNN on transformed features with the aim of providing the first theoretical analysis of kNN over feature transformation.

## 3 Preliminaries

For a training set $\mathcal{D}_n := \{(x_i, y_i)\}_{i \in [n]}$, and a new instance $x$, let $(x_{\pi(1)}, \ldots, x_{\pi(n)})$ be a reordering of the training instances with respect to their distance from $x$, through some appropriate metric. In that setting, the *kNN classifier* $h_{n,k}$ and its *n-sample error rate* are defined by

$$h_{n,k}(x) = \arg\max_{y \in \mathcal{Y}} \sum_{i=1}^{k} \mathbf{1}_{\{y_{\pi(i)} = y\}}, \qquad (R_X)_{n,k} = \mathbb{E}_{X,Y} \mathbf{1}_{\{h_{n,k}(X) \neq Y\}},$$

respectively. The *infinite-sample error rate* of kNN is given by $(R_X)_{\infty,k} = \lim_{n \to \infty}(R_X)_{n,k}$.

*Bayes error rate (BER)*, the error rate of the *Bayes optimal classifier*, often plays a central role in the analysis of kNN classifiers [7, 13, 18]. It is the lowest error rate among all possible classifiers from $\mathcal{X}$ to $\mathcal{Y}$, and can be expressed as

$$R_{X,Y}^* = \mathbb{E}_X[1 - \max_{y \in \mathcal{Y}} \eta_y(x)],$$

which we abbreviate to $R_X^*$ when $Y$ is obvious from the context. Under applying certain transformation $f : \mathcal{X} \to \tilde{\mathcal{X}}$, we denote the BER in $\tilde{\mathcal{X}}$ by $R_{f(X)}^*$, and further define $\Delta_{f,X}^* = R_{f(X)}^* - R_X^*$.

# 4 Impact of Transformations on kNN Convergence

The main goal of this section is to provide a novel theoretical guarantee on the behavior of convergence rates of a kNN classifier over feature transformations. We state the main ingredients and show how they are used to prove the main theorem, whilst the proofs for these ingredients can be found in Section A of the supplementary material, in the same order as presented here.

**Overview of Results.** We begin by providing an overview of our results before expanding them in details in the next two sections.

It is well known that a kNN classifier can converge arbitrarily slowly if one does not assume any regularity [2]. A common starting assumption for determining convergence rates of a kNN classifier is that of the a-posteriori probability $\eta(x)$ being an *L-Lipschitz* function, that is for all $x, x' \in \mathcal{X}$,

$$|\eta(x) - \eta(x')| \leq L\|x - x'\|. \tag{4.1}$$

In order to stay exposition friendly, we will stay close to the Lipschitz condition by imposing the mildest assumptions, as found in [18]. We assume that we are given the *raw* data in $\mathcal{X} \times \mathcal{Y}$, where $\mathcal{X} \subset \mathbb{R}^D$. Our reference point will be Theorem 6.2 from [18] which states that if $\mathcal{X}$ is bounded and $\eta(x)$ is $L$-Lipschitz, then

$$\mathbb{E}_n\left[(R_X)_{n,k}\right] - R_X^* = \mathcal{O}\left(\frac{1}{\sqrt{k}}\right) + \mathcal{O}\left(L\left(\frac{k}{n}\right)^{1/D}\right). \tag{4.2}$$

The main result of this section is an extension of the above to *transformed* data.

**Theorem 4.1.** *Let $\mathcal{X} \subseteq \mathbb{R}^D$ and $\tilde{\mathcal{X}} \subseteq \mathbb{R}^d$ be bounded sets, and let $(X, Y)$ be a random vector taking values in $\mathcal{X} \times \{0, 1\}$. Let $g: \tilde{\mathcal{X}} \to \mathbb{R}$ be an $L_g$-Lipschitz function. Then for all transformations $f: \mathcal{X} \to \tilde{\mathcal{X}}$, one has*

$$\mathbb{E}_n\left[(R_{f(X)})_{n,k}\right] - R_X^* = \mathcal{O}\left(\frac{1}{\sqrt{k}}\right) + \mathcal{O}\left(L_g\left(\frac{k}{n}\right)^{1/d}\right) + \mathcal{O}\left(\sqrt[4]{\mathcal{L}_{g,X}(f)}\right). \tag{4.3}$$

**Implications.** An obvious comparison of (4.2) and (4.3) shows that applying a transformation can introduce a non-negligible bias, and thus a feature transformation should not be used if one has an infinite pool of data points. However, as we will confirm in the experimental section, in the finite-sample regime the benefit of reducing the dimension and, often more importantly, changing the geometry of the space using (pre-trained) transformations significantly outweighs the loss introduced through this bias. In practice, $g$ is typically chosen to be a linear layer with the sigmoid output function, that is $g_w(x) = \sigma(\langle w, x \rangle)$, where $\sigma(x) := (1 + e^{-x})^{-1}$. It is easy to see that $\sigma$ is $1/4$-Lipschitz since $d\sigma/dx = \sigma(1 - \sigma) \leq 1/4$, whereas $x \mapsto \langle w, x \rangle$ is $\|w\|_2$-Lipschitz by the Cauchy-Schwarz inequality, implying that $g_w$ is $\|w\|_2/4$-Lipschitz. Therefore, for any $w$ we can simply insert $\mathcal{L}_{g_w,X}$ and $\|w\|_2$, since they comprise a valid bound in (4.3). In particular, this aligns with a common heuristic of minimizing a loss, while introducing the additional task of capturing $\|w\|_2$ in the process.

The proof of Theorem 4.1 is divided into two parts, motivated by

$$\mathbb{E}_n\left[(R_{f(X)})_{n,k}\right] - R_X^* = \underbrace{\mathbb{E}_n\left[(R_{f(X)})_{n,k}\right] - R_{f(X)}^*}_{\text{convergence rates of vanilla kNN on } f(X)} + \underbrace{R_{f(X)}^* - R_X^*}_{\text{safety of } f}. \tag{4.4}$$

We examine these parts in the next two sections.

## 4.1 Safe Transformations

We start by bounding the increase in the Bayes error that a transformation can introduce. In order to motivate what follows, note that one can rewrite $\Delta_{f,X}^*$ as[2]

$$\Delta_{f,X}^* = \mathbb{E}_{x \sim X}\left[p(y_x \mid x) - p_{f^{-1}}(y_{f(x)} \mid f(x))\right], \tag{4.5}$$

where $y_x = \arg\max_{y \in \mathcal{Y}} p(y|x)$ and $y_{f(x)} = \arg\max_{y \in \mathcal{Y}} p_{f^{-1}}(y|f(x))$. It is clear that any feature transformation can only increase the Bayes error (as seen in Figure 2a), due to the fact that $\max(.)$ is a convex function. Hence, in order to understand and control this increase, we define and analyze *safe* transformations – those which increase the Bayes error only for a small value.

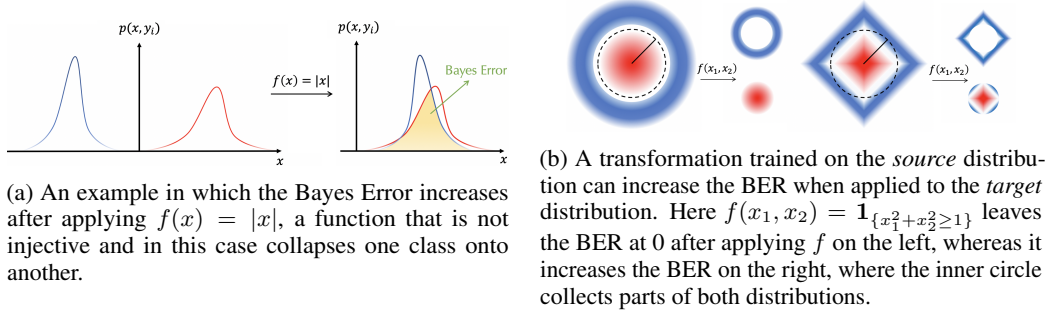

(a) An example in which the Bayes Error increases after applying $f(x) = |x|$, a function that is not injective and in this case collapses one class onto another.

(b) A transformation trained on the *source* distribution can increase the BER when applied to the *target* distribution. Here $f(x_1, x_2) = \mathbf{1}_{\{x_1^2 + x_2^2 \geq 1\}}$ leaves the BER at 0 after applying $f$ on the left, whereas it increases the BER on the right, where the inner circle collects parts of both distributions.

Figure 2: Transformations introduce bias by increasing the BER.

**Definition 4.2.** *We say that a transformation $f \colon \mathcal{X} \to \tilde{\mathcal{X}}$ is $\delta$-safe (with respect to $p$) if $\Delta_{f,X}^* \leq \delta$, and $\delta$-unsafe (with respect to $p$) if there is no $\delta' < \delta$ such that $\Delta_{f,X}^* \leq \delta'$.*
*When $\delta = 0$ we simply say that a function is* safe *or* unsafe.

Motivated by (4.5), we see that any injective function $f$ is safe, since for all $x \in \mathcal{X}$ one has $f^{-1}(\{f(x)\}) = \{x\}$. For example, this implies that $x \mapsto (x, f(x))$ is a safe transformation, for any map $f$. In the supplementary material we weaken the notion of injectivity, allowing us to deduce that the CReLU (Concatenated ReLU) activation function [31], defined as $x \mapsto (x^+, x^-)$, where $x^+ = \max\{x, 0\}$ and $x^- = \max\{-x, 0\}$, is a safe function. Interestingly, both $x^+$ and $x^-$ are $1/2$-unsafe, showing that unsafe functions can be concatenated into a safe one.

When discussing safety using injective functions, one is restricted to only considering how $f$ behaves on $\mathcal{X}$, without using the information that $\mathcal{Y}$ might facilitate. For example, a function that reduces the dimension is not injective, whilst in reality we might be able to avoid the loss of information that $\mathcal{Y}$ carries about $\mathcal{X}$, when it comes to the Bayes error. A perfect such example is the map $f_{\mathcal{Y}} \colon \mathcal{X} \to \mathcal{Y}$ defined by $f_{\mathcal{Y}}(x) := y_x$, as $y_{f_{\mathcal{Y}}(x)} = \arg\max_{y \in \mathcal{Y}} p_{f_{\mathcal{Y}}^{-1}}(y|f_{\mathcal{Y}}(x)) = \arg\max_{y \in \mathcal{Y}} p_{f_{\mathcal{Y}}^{-1}}(y|y_x) = y_x$. Through (4.5) we see that $f_{\mathcal{Y}}$ does not change the Bayes error even though it reduces the size (of $\tilde{\mathcal{X}}$) to the number of classes. Thus, we now provide an alternative sufficient condition for $\delta$-safe functions, which takes into account both $\mathcal{X}$ and $\mathcal{Y}$ through the *Kullback-Leibler divergence*[3], given by

$$D_{KL}(q_1(x) \| q_2(x)) := \sum_{x \in \mathcal{X}} q_1(x) \log \frac{q_1(x)}{q_2(x)}.$$

We further denote $D_{KL}(p(y|x) \| p_{f^{-1}}(y|f(x))) := D_{KL}\left(p(x,y) \| \frac{p(x)}{p_{f^{-1}}(f(x))} p_{f^{-1}}(f(x), y)\right)$, which one can think of as the loss in the *mutual information* after applying $f$.

**Lemma 4.3.** *Let $\mathcal{X}$ and $\mathcal{Y}$ be finite sets, and let $f \colon \mathcal{X} \to \tilde{\mathcal{X}}$ be a transformation such that $D_{KL}(p(y|x) \| p_{f^{-1}}(y|f(x))) \leq (2/\ln 2)\delta^2$. Then $f$ is $\delta$-safe.*

We remark that the converse of Lemma 4.3 does not hold, meaning that one can have a $\delta$-safe function for which the corresponding KL divergence is larger than $(2/\ln 2)\delta^2$. For example, the above mentioned $f_{\mathcal{Y}}$ is a safe transformation, whereas the KL-divergence is strictly positive as soon as $p(y|x) \neq p_{f^{-1}}(y|f(x))$, for some $x$ and $y$, which is easy to construct. On the other hand, we can prove that the bounds are tight in the following sense.

**Lemma 4.4.** *Let $\mathcal{X}, \tilde{\mathcal{X}}$ be finite sets satisfying $|\tilde{\mathcal{X}}| < |\mathcal{X}|$, $\mathcal{Y} = \{0, 1\}$, and let $f \colon \mathcal{X} \to \tilde{\mathcal{X}}$ be a transformation. For any $\delta \in [0, 1/2)$ there exist random variables $X, Y$, taking values in $\mathcal{X}, \mathcal{Y}$, such that $R_{f(X)}^* - R_X^* = \delta$ and $D_{KL}(p(y|x) \| p_{f^{-1}}(y|f(x))) = (2/\ln 2)\delta^2 + O(\delta^4)$.*

A key challenge with the proposed approach is that the feature transformation might have been trained on a significantly different joint probability distribution, and, as such, might change the Bayes error in an unfavourable way when applied to our distribution of interest (Figure 2b). It is clear that in the worst case this can induce an unbounded error in the estimation. In practice, however, these feature transformations are learned for each domain specifically (e.g. deep convolutional neural networks for the vision domain, transformers for NLP tasks), and recent results in transfer learning show that such transformations do indeed *transfer* to a variety of downstream tasks [43]. Guided by this intuition we derive a sufficient condition under which our notion of safety is preserved on similar distributions.

**Theorem 4.5.** *Let $p_S$ and $p_T$ be two probability distributions defined on $\mathcal{X} \times \mathcal{Y}$ that satisfy $D_{KL}(p_S \,\|\, p_T) \leq \varepsilon^2/(8 \ln 2)$. If a transformation $f$ is $\delta$-safe with respect to $p_S$, then $f$ is $(\delta + \varepsilon)$-safe with respect to $p_T$.*

From (4.2) we know that a kNN classifier on $f(X)$ converges to $R^*_{f(X)}$ under some mild assumptions. Since one aims at solving the original task, in other words to examine the convergence of $(R_{f(X)})_{n,k}$ to $R^*_X$, we need to make the transition to $R^*_X$ by using some form of $\mathcal{L}_{g,X}(f)$. We conclude this section by stating the theorem that allows this transition.

**Theorem 4.6.** *For every transformation $f \colon \mathcal{X} \to \tilde{\mathcal{X}}$, one has $\Delta^*_{f,X} \leq 2\sqrt{\mathcal{L}_{g,X}(f)}$.*

## 4.2 Convergence Rates of a kNN Classifier over Transformed Features

In this section we derive convergence rates over transformed features, which together with Theorem 4.6 yield the proof of Theorem 4.1, via (4.4). We provide statements for $X$, i.e. the raw data, keeping in mind that we apply the main theorem of this section on $f(X)$ later on. In order to get a convergence rate that works on $f(X)$ without imposing any particular structure on $f$, one needs a weaker notion than the Lipschitz condition. It suffices to define the *probabilistic Lipschitz condition*.

**Definition 4.7.** *Let $\varepsilon, \delta, L > 0$, and let $X, X' \in \mathcal{X}$ be random variables sampled using the same joint probability distribution $p$ on $\mathcal{X} \times \{0,1\}$. We say that $\eta(x) = p(1|x)$ is $(\varepsilon, \delta, L)$-probably Lipschitz if*
$$\mathbb{P}\Big(|\eta(X) - \eta(X')| \leq \varepsilon + L\|X - X'\|\Big) \geq 1 - \delta.$$

With such a condition, similar to the one of Theorem 6.2 in [18], we can prove the following result.

**Theorem 4.8.** *Let $\mathcal{X} \subseteq \mathbb{R}^d$ be a bounded set, $X \in \mathcal{X}$, $Y \in \{0,1\}$. If $\eta(x)$ is $(\varepsilon, \delta, L)$-probably Lipschitz, then*
$$\mathbb{E}_n\left[(R_X)_{n,k}\right] - R^*_X = \mathcal{O}\left(\frac{1}{\sqrt{k}}\right) + \mathcal{O}\left(L\left(\frac{k}{n}\right)^{1/d}\right) + \mathcal{O}\left(\sqrt{\delta} + \varepsilon\right). \qquad (4.6)$$

The first term represents the variance in the training set, the second term comes from the Lipschitz-like structure of $X$, whilst the last term comes from the need of the probabilistic Lipschitz condition, which will later allow us to avoid additional constraints on $f$. We use Theorem 4.8 on $f(X)$, which needs an evidence of probably Lipschitz condition with respect to the $g$-squared loss. The following consequence of Markov's inequality yields the desired evidence. Since it is applied on $f(X)$, it suffices to have the identity function in place of $f$.

**Lemma 4.9.** *Let $g$ be an $L$-Lipschitz function. Then for any real number $\varepsilon > 0$, the function $\eta(x)$ is $(\varepsilon, 8\mathcal{L}_{g,X}(id)/\varepsilon^2, L)$-probably Lipschitz.*

**Proof sketch of the main result.** Deducing Theorem 4.1 is now straightforward – Lemma 4.9 says that we can apply Theorem 4.8 with $X$ substituted by $f(X)$, meaning that for any $f$ and any $\varepsilon > 0$,
$$\mathbb{E}_n\left[(R_{f(X)})_{n,k}\right] - R^*_{f(X)} = \mathcal{O}\left(\frac{1}{\sqrt{k}}\right) + \mathcal{O}\left(L\left(\frac{k}{n}\right)^{1/d}\right) + \mathcal{O}\left(\frac{\sqrt{\mathcal{L}_{g,f(X)}(id)}}{\varepsilon} + \varepsilon\right). \quad (4.7)$$

We optimize this by setting $\varepsilon = \sqrt[4]{\mathcal{L}_{g,f(X)}(id)}$. In the supplementary material, we prove an easily attainable upper bound $\mathcal{L}_{g,f(X)}(id) \leq \mathcal{L}_{g,X}(f)$, which, together with the final transition from $R^*_{f(X)}$ to $R^*_X$ that is available via Theorem 4.6, proves the main result.

**Discussions.** We could further optimize the exponent of $\mathcal{L}_{g,X}$, however, we leave this for future work, as it already serves our purpose of providing a connection between the $g$-squared loss of $f$ and the Lipschitz constant of $g$. Furthermore, one might argue that (4.2) could be used to yield a better rate of convergence than the one in (4.3), provided that the Lipschitz constant of $\eta_{f^{-1}}$ is known for every $f$. However, checking whether $\eta_{f^{-1}}$ is $L$-Lipschitz is not a feasible task. For example, computing the best Lipschitz constant of a deep neural network is an NP-hard problem [29], whilst upper bounding the Lipschitz constant over high-dimensional real-world data and many transformations becomes impractical. Our result provides a practical framework that can easily be deployed for studying the influence of feature transformations on a kNN classifier, as illustrated in the next section.

Table 1: Dataset Statistics

| NAME | DIMENSION | CLASSES | TRAINING SAMPLES | TEST SAMPLES |
|------|-----------|---------|------------------|--------------|
| MNIST | 784 | 10 | 60K | 10K |
| CIFAR10 | 3072 | 10 | 50K | 10K |
| CIFAR100 | 3072 | 100 | 50K | 10K |
| IMDB | 104083 | 2 | 25K | 25K |
| SST2 | 14583 | 2 | 67K | 872 |
| YELP | 175710 | 5 | 500K | 50K |

## 5 Experimental Results

The goal of this section is to show that one can use the upper bounds on the convergence rate of kNN, derived in the previous sections, to explain the impact of different feature transformations on the kNN performance. This works particularly well when compared with simpler metrics mentioned in the introduction, such as the dimension or the accuracy of another trained classifier on the same space.

In order to empirically verify our theoretical results, as discussed in the implications of Theorem 4.1, we focus on the logistic regression model, since it is most commonly used in practice, and for which one can easily calculate the Lipschitz constant. More precisely, we examine $g(x) = \sigma(\langle w, x \rangle)$, with $L_g = \|w\|_2$. When dealing with multi-class tasks, we use

$$g(x) = \text{softmax}\left(W^T x + b\right),$$

whilst reporting $\|W\|_F$, the Frobenius norm of the weights, in place of $\|w\|_2$.

Calculating the $g$-squared loss of $f$ is not feasible in practice, since one usually does not know the true distribution $\eta(X)$. However, using the fact that $\mathbb{E}_{Y|X=x}(\eta(x) - Y) = 0$, one can decompose the loss as

$$\mathcal{L}_{g,X}(f) = \mathbb{E}_{X,Y}\left((g \circ f)(X) - Y\right)^2 + \mathbb{E}_{X,Y}\left(\eta(X) - Y\right)^2.$$

As the second term does not depend on $g$ or $f$, we can rank feature transformations by estimating the first term for a fixed $g$. With that in mind, when we compare different transformations, we report the *mean squared error*[4] of the test set, defined by

$$MSE_g(f, W, b) = \frac{1}{|TEST|} \sum_{i \in TEST} (\text{softmax}(W^T f(x_i) + b) - y_i)^2.$$

For kNN, we restrict ourselves to the Euclidean distance, the most commonly used distance function. Furthermore, we set $k = 1$ for this entire section, whereas the empirical analysis of the influence of $k > 1$ is conducted in Section B of the supplementary material.

### 5.1 Datasets

We perform the evaluation on two data modalities which are ubiquotous in modern machine learning. The first group consists of *visual classification tasks*, including MNIST, CIFAR10 and CIFAR100. The second group consists of standard *text classification tasks*, where we focus on IMDB, SST2 and YELP. The details are presented in Table 1. We remark that for the visual classification tasks, the raw features are the pixel intensities, whereas for the text classification we apply the standard bag-of-words preprocessing, with and without term-frequency/inverse-document-frequency weighting [22].

### 5.2 Feature Transformations

We run the experiments on a diverse set of feature transformations used in practice. The list of 30 transformations includes standard dimension reduction methods like the Principal Component Analysis (PCA) and Neighborhood Component Analysis (NCA), as well as using pre-trained feature extraction networks available in TensorFlow Hub and PyTorch Hub. A complete list of the used transformations per dataset is given in the supplementary material.

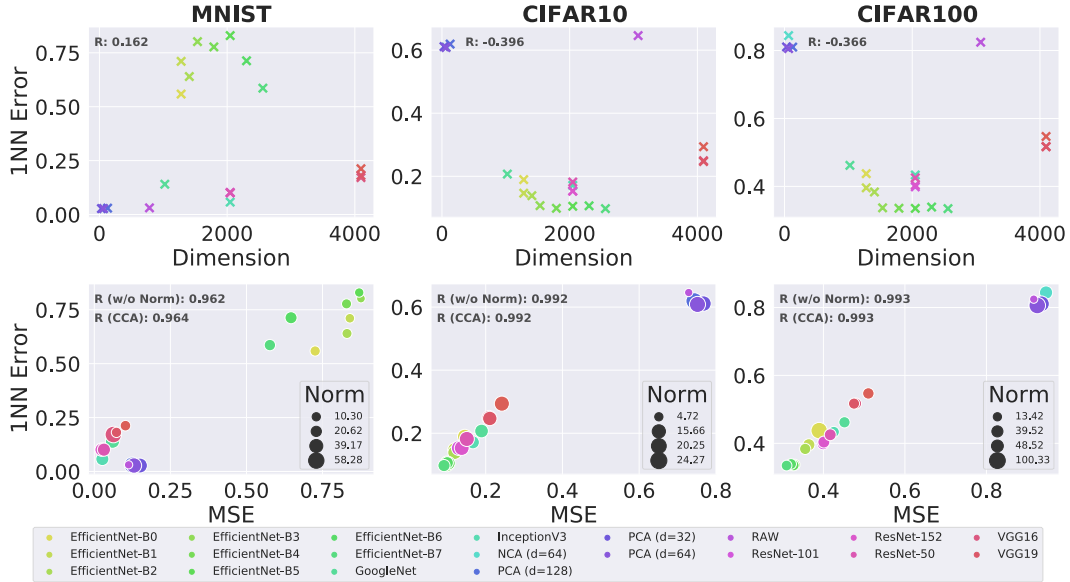

Figure 3: Results on computer vision datasets. **(Top row)** There is a poor linear correlation between the feature dimension and the 1NN error. **(Bottom row)** Comparing the 1NN error with the MSE shows a very high linear correlation. Combining MSE with the norm marginally improves the correlation coefficients.

## 5.3  Results

We use *Pearson's r correlation* [5] to analyze different approaches for comparing feature transformations with respect to the kNN error. Pearson's r ranges from -1 to 1, with 1 implying that two random variables have a perfect positive (linear) correlation, where -1 implies a perfect negative correlation. We provide an extended experimental evaluation including convergence plots and a detailed description of the experimental setting (such as the protocol used for training $W$) in Section B of the supplementary material, whereas the source code for reproducing the numbers can be found in the supplementary files.

**Feature Dimension.**  We start by inspecting the dimension of the resulting feature space and its achieved kNN error. We exhibit a very poor (or even negative) correlation on all text datasets, as illustrated in the top rows of Figures 3 and 4.

**MSE vs 1NN Error.**  On computer vision tasks we observe high correlation when comparing the 1NN error with the MSE, with marginal improvements when the norm is included, as reported in the bottom row of Figure 3. This is in line with (4.3) by simply concluding that the last term dominates.

On text classification datasets the MSE alone struggles to explain the performance of 1NN on top of feature transformations, as visible in the bottom of Figure 4. Without considering the norm (i.e., ignoring the size of the circles on Figure 4), we see a number of misjudged (important) examples, despite having a positive correlation.

**Validation Based on Our Theoretical Results – MSE and Norm.**  In order to report the correlation between multiple variables (notably the 1NN error vs. the combination of MSE and the norm), we first reduce the dimension of the latter by performing a canonical-correlation analysis (CCA) [35] which enables us to evaluate Pearson's r afterwards.

As visible in the second row of Figure 4, there is a significant improvement in correlation on all the text classification datasets. Furthermore, this method enables us to separate transformations that achieve similar MSE, but with different norms, in particular on those transformations that perform well. We can see that in those cases, the smaller norm implies smaller 1NN error. Including the norm into the analysis of computer vision datasets, we see some marginal improvements, although on already high correlation values, which is also aligned with our theoretical analysis.

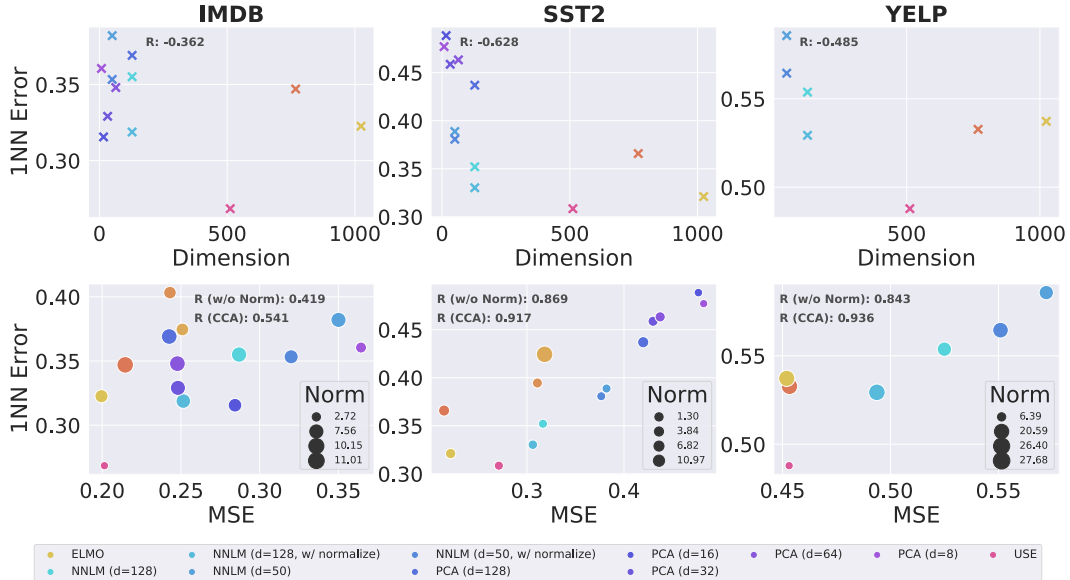

Figure 4: Results on text classification datasets. **(Top row)** There is a poor linear correlation between the feature dimension and the 1NN error. **(Bottom row)** Comparing the 1NN error with MSE without the norm (i.e., by ignoring the sizes of the circles) leads a strong positive correlation. Including the norm improves this further, particularly on important cases (the bottom left corner) which makes the inclusion of norm significant.

**Empirical Results for $k > 1$.** In addition to the experiments presented in the previous section, we performed experiments for $k > 1$ over all datasets and all embeddings.[5] All the results for $k > 1$ confirm the theoretical findings, whilst for some $k$ they offer an improvement in the linear correlation (e.g., on SST2 one can get CCA score up to 0.97 when $k = 8$, whereas in Figure 4 we report 0.917, for $k = 1$). In order to simplify the exposition, we opted for $k = 1$ and $MSE$ instead of $MSE^{1/4}$ since it already serves the purpose of showing that $MSE$ is important on its own, whereas including the smoothness assumption through the norm further improves the correlation.

## 6  Final remarks

One could examine the tightness of our theoretical bounds by constructing a toy dataset with a known true posterior probability and a set of transformations. However, any such transformation should either be an identity (if the toy dataset already has the desired property), or carefully constructed for this purpose only. In this work we opted for not constructing a single transformation ourselves, as our main goal is to bridge the gap between the real-world applications (e.g. pre-trained embeddings) with the theory of nearest neighbors. For example, even Lemma 4.4, which establishes the sharpness of the safety bound, works for any transformation. For that purpose, our probabilistic Lipschitz condition is as weak as possible. To this end, studying CCA of $k^{-1/2}$, $\|w\|(k/n)^{1/d}$ and $MSE^{1/4}$ (or some other exponent under stronger assumptions) for $k > 1$, could provide another angle for understanding the tightness of the bound, which is out of the scope of this work.

**Conclusion.** The goal of this work is to provide novel theoretical results aimed towards understanding the convergence behavior of kNN classifiers over transformations, bridging the existing gap between the current state of the theory and kNN applications used in practice. We provide a rigorous analysis of properties of transformations that are highly correlated with the performance of kNN classifiers, yielding explainable results of kNN classifiers over transformations. We believe that optimizing the upper bound presented here and extending results to classifiers other than logistic regression could form an interesting line of future research.

## Acknowledgements

We are grateful to Mario Lucic and André Susano Pinto for their constructive feedback. CZ and the DS3Lab gratefully acknowledge the support from the Swiss National Science Foundation (Project Number 200021_184628), Innosuisse/SNF BRIDGE Discovery (Project Number 40B2-0_187132), European Union Horizon 2020 Research and Innovation Programme (DAPHNE, 957407), Botnar Research Centre for Child Health, Swiss Data Science Center, Alibaba, Cisco, eBay, Google Focused Research Awards, Oracle Labs, Swisscom, Zurich Insurance, Chinese Scholarship Council, and the Department of Computer Science at ETH Zurich.

## Broader Impact

One of the current bottlenecks in machine learning is the lack of robustness and explainability. It is well known that kNN has properties (some of them cited in the introduction) that allow one to tackle these challenges. However, when it comes to accuracy, kNN on its own is often inferior to modern day machine learning methods, limiting the possible impact. In this paper we propose a novel theoretical framework aimed at understanding the best practices for employing kNN on top of pre-trained feature transformations, in order to gain on all positive aspects of kNN. In particular, we show that by using resources that are already widely available (i.e., open-sourced pre-trained embeddings), without the need of training from scratch, one can improve this already efficient, robust and interpretable classifier. We do not expect any direct negative impact from this work as we purely focus on the theoretical understanding of a classifier that itself is non-controversial.

## Footnotes

[2]When a deterministic feature transformation $f: \mathcal{X} \to \tilde{\mathcal{X}}$ is applied, we define the induced joint probability by $p_{f^{-1}}(\tilde{x}, y) = p(X \in f^{-1}(\{\tilde{x}\}), Y = y)$, where $f^{-1}(\{\tilde{x}\}) = \{x \in \mathcal{X}: f(x) = \tilde{x}\}$ is the preimage of $\tilde{x}$.

[3]Without loss of generality, we assume that $\mathcal{X}$ is finite and logarithm to the base 2.

[4]In these terms, when one wants to rank different classifiers, the MSE is often referred to as the *Brier score* (we refer an interested reader to [6, 8]).

[5]In the supplementary code under ''`code/results/<DATASET>/knn_accuracy/<EMBEDDING>.csv`''

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
