[Supplementary Material 1 · supplementary.pdf]

# A Proofs

## A.1 Safe Transformations

In this section we prove the claims from Section 4.1, in the same order as stated there. Even though the main part of the paper is considering $\mathcal{Y} = \{0, 1\}$ only, which is the usual setting when convergence rates of a kNN classifier are discussed, we suppose that $\mathcal{Y} = \{1, \ldots, C\}$, for some integer $C \geq 2$, until Section A.1.1.

### Safe Transformations via Injectivity

We discuss injective functions in Section 4.1, and provide examples of safe transformations that arise from injectivity, such as $x \mapsto (x, f(x))$, for any map $f$, and $x \mapsto (x^+, x^-)$. We will now prove this by providing a sufficient condition for a function to be $\delta$-safe. We call this condition $\delta$-*injectivity*.

**Definition A.1.** *Let $(\mathcal{X}, \mathcal{A}, p)$ and be a finite probability space, and let $(\tilde{\mathcal{X}}, \tilde{\mathcal{A}})$ be a finite measurable space. We say that a measurable function $f \colon \mathcal{X} \to \tilde{\mathcal{X}}$ is $\delta$-injective if there exists a subset $I_{\mathcal{X}}(f) \subseteq \mathcal{X}$ on which $f$ is injective, and that satisfies*

$$p(I_{\mathcal{X}}(f)) \geq 1 - \delta.$$

**Lemma A.2.** *Let $f_i \colon \mathcal{X} \to \tilde{\mathcal{X}}_i$, for $i = 0, \ldots, n$, be functions such that there exist $I_{\mathcal{X}}(f_0), \ldots, I_{\mathcal{X}}(f_n) \subseteq \mathcal{X}$, sets on which $f_0, \ldots, f_n$ are injective, respectively, and such that*

$$p\left(\bigcup_{i=0}^{n} I_{\mathcal{X}}(f_i)\right) \geq (1 - \delta).$$

*Then $(f_0, \ldots, f_n) \colon \mathcal{X} \to \prod_{i=0}^{n} \tilde{\mathcal{X}}_i$ is $\delta$-safe.*

*In particular, if $f_0$ is $\delta$-injective, then $(f_0, f_1, \ldots, f_n)$ is $\delta$-safe.*

PROOF: We first prove the claim for $n = 0$ and then extend it to $n \in \mathbb{N}$.

(1) $n = 0$:

In this case, we ought to prove that if $f$ is $\delta$-injective, then $f$ is $\delta$-safe. Let $I_{\mathcal{X}}(f)$ be a set on which $f$ is injective and that satisfies $p(I_{\mathcal{X}}(f)) \geq 1 - \delta$. Motivated by (4.5), we define $\mathcal{X}_l := \{x \in \mathcal{X} \colon y_x \neq y_{f(x)}\}$. With the above definition, note that (4.5) can be reduced to

$$\Delta_{f,X}^* = \mathbb{E}_{x \sim X}\left[\left(p(y_x|x) - p_{f^{-1}}(y_{f(x)}|f(x))\right) \cdot \mathbb{1}\{x \in \mathcal{X}_l\}\right] \leq p(\mathcal{X}_l). \tag{A.1}$$

We will now modify $I_{\mathcal{X}}(f)$ to get $I_{\mathcal{X}_l}$ that is of the same mass, and is disjoint from $\mathcal{X}_l$. If $\mathcal{X}_l \cap I_{\mathcal{X}}(f) = \emptyset$, then we are done. Therefore, let $x_l \in \mathcal{X}_l \cap I_{\mathcal{X}}(f)$, implying $y_{x_l} \neq y_{f(x_l)}$. Note that there has to exist $x \in f^{-1}(\{f(x_l)\})$ such that $y_x = y_{f(x_l)} = y_{f(x)}$, as otherwise $y_{f(x)}$ would not be the winning $y$ for $f(x)$. We place $x$ into $I_{\mathcal{X}_l}$, noting that $x \notin \mathcal{X}_l$, and repeat this for every element in $\mathcal{X}_l \cap I_{\mathcal{X}}(f)$. Finally, we add to $I_{\mathcal{X}_l}$ all the elements that are in $I_{\mathcal{X}}(f) \setminus \mathcal{X}_l$. By the construction we see that $I_{\mathcal{X}_l}$ is a set on which $f$ is injective, since we always choose only one representative from each $f^{-1}(\tilde{x})$, and

$$p(I_{\mathcal{X}_l}) = p(\mathcal{X}_l \sqcup (I_{\mathcal{X}}(f) \setminus \mathcal{X}_l)) = p(I_{\mathcal{X}}(f)) \geq 1 - \delta,$$

where $\sqcup$ denotes a disjoint union. Since $\mathcal{X}_l$ and $I_{\mathcal{X}_l}$ are disjoint, this yields $p(\mathcal{X}_l) < \delta$, which together with (A.1) finishes the proof.

(2) $n > 0$:

Let $I_{\mathcal{X}}(f_0, \ldots, f_n) := \bigcup_{i=0}^{n} I_{\mathcal{X}}(f_i)$. It suffices to prove that $(f_0, \ldots, f_n)$ is injective on $I_{\mathcal{X}}(f_0, \ldots, f_n)$, since we already have $p(I_{\mathcal{X}}(f_0, \ldots, f_n)) \geq (1 - \delta)$.

Define $I'_{\mathcal{X}}(f_0), \ldots, I'_{\mathcal{X}}(f_n)$ inductively by $I'_{\mathcal{X}}(f_0) := I_X(f_0)$, and

$$I'_{\mathcal{X}}(f_k) := I_{\mathcal{X}}(f_k) \setminus \left(\bigcup_{j=0}^{k-1} I_{\mathcal{X}}(f_j)\right), \quad k = 1, \ldots, n.$$

Then

$$I_{\mathcal{X}}(f_0, \ldots, f_n) = \bigcup_{i=0}^{n} I_{\mathcal{X}}(f_i) = \bigsqcup_{i=0}^{n} I'_{\mathcal{X}}(f_i).$$

Therefore, it suffices to prove that $(f_0, \ldots, f_n)$ is injective on $\bigsqcup_{i=0}^{n} I'_{\mathcal{X}}(f_i)$.

Let $x, \tilde{x} \in \bigsqcup_{i=0}^{n} I'_{\mathcal{X}}(f_i)$, $x \neq \tilde{x}$, and let $k, l$ be such that $x \in I'_{\mathcal{X}}(f_k), \tilde{x} \in I'_{\mathcal{X}}(f_l)$. Then $f_{\max\{k,l\}}(x) \neq f_{\max\{k,l\}}(\tilde{x})$, for which we use that $I'_{\mathcal{X}}(f_{\max\{k,l\}})$ is injective, if $\tilde{x} \in I'_{\mathcal{X}}(f_{\max\{k,l\}})$, or that $I'_{\mathcal{X}}(f_{\max\{k,l\}})$ is disjoint from all the previous ones. This proves that $(f_0, \ldots, f_n)(x) \neq (f_0, \ldots, f_n)(\tilde{x})$, so $(f_0, \ldots, f_n)$ is injective on $\bigsqcup_{i=0}^{n} I'_{\mathcal{X}}(f_i)$.

To finish the proof, we note that if $f_0$ is $\delta$-injective, then

$$p\left(\bigcup_{i=0}^{n} I_{\mathcal{X}}(f_i)\right) \geq p(I_{\mathcal{X}}(f_0)) \geq (1 - \delta),$$

implying that $(f_0, \ldots, f_n)$ is $\delta$-safe, by the results in the previous paragraph. $\qquad\square$

**Safe Transformations via Information Theory**

In this section we prove Lemmas 4.3 and 4.4. The *mutual information* between random variables $X$ and $Y$, taking values in finite sets $\mathcal{X}$ and $\mathcal{Y}$, is defined as

$$I(X;Y) := D_{KL}(p(x,y) \,||\, p(x)p(y)) = \sum_{x \in \mathcal{X}} \sum_{y \in \mathcal{Y}} p(x,y) \log \frac{p(x,y)}{p(x)p(y)},$$

where the logarithm is in base 2. Lemma 4.3 can be understood as the bound on the allowed loss in the mutual information, by noting that

$$I(X;Y) - I(f(X);Y) = D_{KL}(p(x,y) \,||\, p(x)p(y)) - D_{KL}(p_{f^{-1}}(f(x),y) \,||\, p_{f^{-1}}(f(x))p_{f^{-1}}(y))$$

$$= \mathbb{E}_{p(x,y)} \log \frac{p(x,y)}{p(x)p(y)} - \mathbb{E}_{p(x,y)} \log \frac{p_{f^{-1}}(f(x),y)}{p_{f^{-1}}(f(x))p_{f^{-1}}(y)}$$

$$= \mathbb{E}_{p(x,y)} \log \frac{p(y|x)}{p_{f^{-1}}(y|f(x))} = D_{KL}\left(p(y|x) \,||\, p_{f^{-1}}(y|f(x))\right), \qquad \text{(A.2)}$$

since $p_{f^{-1}}(y) = p(y)$. The proof of Lemma 4.3 starts by connecting the change in the Bayes error with the $L^1$-*norm* of the distance between probability distributions, which, in a finite space, equals twice the *total variation distance*. We conclude the proof by applying *Pinsker's inequality*. For a detailed analysis of all of these terms we refer an interested reader to Chapter 11 in [11].

PROOF OF LEMMA 4.3: Note that (4.5) and the definitions of $y_x$ and $y_{f(x)}$ yield

$$\Delta^*_{f,X} = \mathbb{E}_{x \sim X}\left[p(y_x \mid x) - p_{f^{-1}}(y_{f(x)}|f(x))\right]$$

$$= \mathbb{E}_{x \sim X}\left[p(y_x \mid x) - p_{f^{-1}}(y_x|f(x))\right] + \underbrace{\mathbb{E}_{x \sim X}\left[p_{f^{-1}}(y_x|f(x)) - p_{f^{-1}}(y_{f(x)}|f(x))\right]}_{\leq 0}$$

$$\leq \frac{1}{2}\left|\mathbb{E}_{x \sim X}\left[p(y_x \mid x) - p_{f^{-1}}(y_x|f(x))\right]\right| + \frac{1}{2}\left|\mathbb{E}_{x \sim X}\sum_{y \neq y_x}\left[p(y \mid x) - p_{f^{-1}}(y|f(x))\right]\right|$$

$$\leq \frac{1}{2}\mathbb{E}_{x \sim X}\sum_{y \in \mathcal{Y}}\left|p(y \mid x) - p_{f^{-1}}(y \mid f(x))\right|$$

$$= \frac{1}{2}\left\|p(x,y) - \frac{p(x)}{p_{f^{-1}}(f(x))}p_{f^{-1}}(f(x),y)\right\|_1,$$

where we introduced the sum by expanding

$$1 = p(y_x|x) + \sum_{y \neq y_x} p(y|x) = p_{f^{-1}}(y_x|f(x)) + \sum_{y \neq y_x} p_{f^{-1}}(y|f(x)),$$

whilst using the triangle inequality. Pinsker's inequality implies that

$$\left\| p(x,y) - \frac{p(x)}{p_{f^{-1}}(f(x))} p_{f^{-1}}(f(x),y) \right\|_1 \leq \sqrt{(2\ln 2)D_{KL}\left(p(y\mid x) \mid\mid p_{f^{-1}}(y\mid f(x))\right)} \leq 2\delta,$$

finishing the proof after dividing by 2. □

Finally, we provide a construction which shows that the bound in Lemma 4.3 is of the right order.

PROOF OF LEMMA 4.4: Recall that we define $\eta(x) = p(1\mid x)$. We start with $|\mathcal{X}| = 2$, since the general case will be a straightforward extension of it.

Let $\mathcal{X} = \{x_0, x_1\}$ and $\tilde{\mathcal{X}} = \{\tilde{x}\}$. We define $p$ on $\mathcal{X} \times \mathcal{Y}$ by $p(x_0) = p(x_1) = 1/2$, and $\eta(x_0) = \frac{1}{2} - \delta$, $\eta(x_1) = \frac{1}{2} + \delta$, which defines $p(x,y)$. For the change in the Bayes error we have

$$\Delta^*_{f,X} = \frac{1}{2} - \sum_{x \in \{x_0, x_1\}} p(x) \min\{\eta(x), 1 - \eta(x)\} = \delta.$$

For the KL-divergence, note that

$$D_{KL}(p(y\mid x) \mid\mid p_{f^{-1}}(y\mid f(x))) = \sum_{x \in \{x_0, x_1\}} \sum_{y \in \{0,1\}} p(x,y) \log \frac{p(y\mid x)}{p_{f^{-1}}(y\mid f(x))}$$

$$= \sum_{x \in \{x_0, x_1\}} p(x) \sum_{y \in \{0,1\}} p(y\mid x) \log 2p(y\mid x)$$

$$= \sum_{x \in \{x_0, x_1\}} p(x) \left(\eta(x_0) \log 2\eta(x_0) + \eta(x_1) \log 2\eta(x_1)\right)$$

$$= \frac{1}{2} \left((1 - 2\delta)\log(1 - 2\delta) + (1 + 2\delta)\log(1 + 2\delta)\right),$$

where we used $\eta(x_0) = \frac{1}{2} - \delta = 1 - \eta(x_1)$. Taylor expansion for $|x| < 1$ gives

$$(1 + x)\ln(1 + x) + (1 - x)\ln(1 - x) = 2\sum_{k \in \mathbb{N}} \frac{1}{(2k-1)2k} x^{2k},$$

which implies

$$D_{KL}(p(y\mid x) \mid\mid p_{f^{-1}}(y\mid f(x))) = \frac{1}{\ln 2} \sum_{k \in \mathbb{N}} \frac{2^{2k}}{(2k-1)2k} \delta^{2k}$$

$$= \frac{2}{\ln 2}\delta^2 + \frac{4}{3\ln 2}\delta^4 + \frac{32}{15\ln 2}\delta^6 + \ldots = (2/\ln 2)\delta^2 + O(\delta^4),$$

finishing the proof for $|\mathcal{X}| = 2$.

Suppose that $|\mathcal{X}| > 2$. Since $|\tilde{\mathcal{X}}| < |\mathcal{X}|$, we know that there exists a $\tilde{x}$ such that $|f^{-1}(\tilde{x})| \geq 2$, so let $x_0, x_1 \in f^{-1}(\tilde{x})$ be distinct. We define $p$ on $\mathcal{X} \times \mathcal{Y}$ as $p(x,y) = 0$ for $x \notin \{x_0, x_1\}$, while for $p(x_0, y), p(x_1, y)$ we do the above construction, which proves the lemma. □

For $|\mathcal{X}| > 2$ we used the most simple construction, however, one can extend the idea behind the proof for $|\mathcal{X}| = 2$ into a more general one. For example, we can define a probability distribution in which for all $x \in \mathcal{X}$ one has $\eta(x) \in \{\frac{1}{2} - \delta, \frac{1}{2} + \delta\}$, with the same proportion of each. In other words, each $x$ is a bucket with either $\frac{1}{2} - \delta$ values being 1, or $\frac{1}{2} + \delta$ values being 1. Let

$$\mathcal{X}_0 := \left\{x \in \mathcal{X} : \eta(x) = \frac{1}{2} - \delta\right\}, \qquad \mathcal{X}_1 := \left\{x \in \mathcal{X} : \eta(x) = \frac{1}{2} + \delta\right\},$$

thus $\mathcal{X} = \mathcal{X}_0 \sqcup \mathcal{X}_1$. Now $f$ can either merge buckets of the same type, in which case neither do the Bayes error nor the KL-divergence change, or buckets of a different type, where the changes are $\delta$ and $2\delta^2 + O(\delta^4)$, respectively. Choosing the right proportion of each bucket in $f^{-1}(\tilde{x})$ is now an easy task, yielding the construction.

**Safe Transformations on Similar Probability Distributions**

In this section we prove Theorem 4.5. As mentioned in the main body, transformations used for estimating the Bayes error might have been trained on a distribution different then the target one, and as such, might change the Bayes error in an unfavourable way when applied to the distribution of interest. We investigate this in the next few paragraphs.

Let $p_S(x, y)$ be the *source* probability distribution based on random variables $X_S \in \mathcal{X}_S$ and $Y_S \in \mathcal{Y}_S$, which is the probability distributions used for training a transformation $f_S$. With $p_T(x, y)$ we denote the *target* probability distribution, the one that serves as the basis for random variables $X_T \in \mathcal{X}_T$ and $Y_T \in \mathcal{Y}_T$. Theorem 4.5 provides a sufficient condition on the relationship between $p_S$ and $p_T$, in terms of the Kullback-Leibler divergence, so that a $\delta$-safe transformation with respect to $p_S$ is a $\delta'$-safe transformation with respect to $p_T$.

Before we start with the proof, let us argue why it makes sense to set $\mathcal{X}_S = \mathcal{X}_T = \mathcal{X}$ and $\mathcal{Y}_S = \mathcal{Y}_T = \mathcal{Y}$, as it is assumed in Theorem 4.5, even when we have more then two classes. When it comes to $\mathcal{X}$, any pre-trained feature transformation comes with a fixed input dimension. Therefore, in order to apply a feature transformation one usually needs to modify the input vector. When dealing with images, this often means resizing the image, whether it is by scaling the image, or by adding white/black pixels. This is an injective process as long as we do not reduce the dimension, which is reasonable to assume as we usually use transformations trained on larger inputs. Therefore, instead of $\mathcal{X}_S$ we can consider a probability distribution mapped through an injective map $g\colon \mathcal{X}_S \to \mathcal{X}$, which is a safe transformation. We will omit the mention of $g$ for the ease of notation. For $\mathcal{Y}$, we first assume that $\mathcal{Y}_T \subseteq \mathcal{Y}_S$, since we want to use feature transformations that work well on more difficult tasks. When $f_S$ is safe with respect to $p_S$ on $\mathcal{X}_S \times \mathcal{Y}_S$, it is easy to see that $f_S$ is also safe with respect to the restriction of $p_S$ to $\mathcal{X}_S \times \mathcal{Y}_T$. This does not necessarily hold when we weaken the condition to $\delta$-safe. In that case, our assumption is that $f$ is $\delta$-safe with respect to $p_S$ on $\mathcal{X}_S \times \mathcal{Y}_T$ in the first place, thus taking $\mathcal{Y}_T$ as the source $\mathcal{Y}$.

PROOF OF THEOREM 4.5: Note that

$$R^*_{f(X_T)} - R^*_{X_T} \leq \underbrace{\left| R^*_{f(X_T)} - R^*_{f(X_S)} \right|}_{I_1} + \underbrace{\left| R^*_{f(X_S)} - R^*_{X_S} \right|}_{I_2} + \underbrace{\left| R^*_{X_S} - R^*_{X_T} \right|}_{I_3}.$$

Since $f$ is $\delta$-safe with respect to $p_S$, we have $I_2 \leq \delta$.

For $I_1$, let $\tilde{p}_S := p^{(S)}_{f^{-1}}$ and $\tilde{p}_T := p^{(T)}_{f^{-1}}$ denote the corresponding measures with respect to $\tilde{\mathcal{X}}$, and let

$$y^{(S)}_{\tilde{x}} = \arg\max_{y \in \mathcal{Y}} \tilde{p}_S(\tilde{x}, y), \quad y^{(T)}_{\tilde{x}} = \arg\max_{y \in \mathcal{Y}} \tilde{p}_T(\tilde{x}, y).$$

For a fixed $\tilde{x}$ we can assume without loss of generality that $\tilde{p}_S(\tilde{x}, y^{(S)}_{\tilde{x}}) \geq \tilde{p}_T(\tilde{x}, y^{(T)}_{\tilde{x}})$. Then

$$\left| \max_{y \in \mathcal{Y}} \tilde{p}_S(\tilde{x}, y) - \max_{y \in \mathcal{Y}} \tilde{p}_T(\tilde{x}, y) \right| = \tilde{p}_S(\tilde{x}, y^{(S)}_{\tilde{x}}) - \tilde{p}_T(\tilde{x}, y^{(T)}_{\tilde{x}})$$

$$\leq \tilde{p}_S(\tilde{x}, y^{(S)}_{\tilde{x}}) - \tilde{p}_T(\tilde{x}, y^{(S)}_{\tilde{x}})$$

$$\leq \sum_{y \in \mathcal{Y}} \left| \tilde{p}_S(\tilde{x}, y) - \tilde{p}_T(\tilde{x}, y) \right|.$$

Summing the above over all $\tilde{x} \in \tilde{\mathcal{X}}$ yields

$$I_1 = \left| \sum_{\tilde{x} \in \tilde{\mathcal{X}}} \left[ \max_{y \in \mathcal{Y}} \tilde{p}_S(\tilde{x}, y) - \max_{y \in \mathcal{Y}} \tilde{p}_T(\tilde{x}, y) \right] \right|$$

$$\leq \sum_{\tilde{x} \in \tilde{\mathcal{X}}} \sum_{y \in \mathcal{Y}} \left| \tilde{p}_S(\tilde{x}, y) - \tilde{p}_T(\tilde{x}, y) \right|$$

$$\overset{\triangle}{\leq} \sum_{\tilde{x} \in \tilde{\mathcal{X}}} \sum_{y \in \mathcal{Y}} \sum_{x \in f^{-1}(\tilde{x})} \left| p_S(x, y) - p_T(x, y) \right|$$

$$= \sum_{x \in \mathcal{X}} \sum_{y \in \mathcal{Y}} \left| p_S(x, y) - p_T(x, y) \right| = \| p_S - p_T \|_1.$$

Repeating the same calculation for $I_3$ implies $I_3 \leq \|p_S - p_T\|_1$. Combining the bounds for $I_1, I_2$ and $I_3$ yields

$$R^*_{f(X_T)} - R^*_{X_T} \leq \delta + 2\|p_S - p_T\|_1.$$

As in the previous section, Pinsker's inequality implies

$$R^*_{f(X_T)} - R^*_{X_T} \leq \delta + 2\sqrt{(2\ln 2)D_{KL}(p_S \,\|\, p_T)} \leq \delta + \varepsilon,$$

concluding the proof. $\qquad\square$

### A.1.1 Safety and the $g$-squared loss

In this section we provide a characterization of $\delta$-safe functions in terms of the $g$-squared loss of $f$, by proving Theorem 4.6. Since this will serve as a connecting point between the rates of convergence of a kNN classifier and the safety of a transformation, from this point onwards we restrict ourselves to binary classification, assuming that $\mathcal{Y} = \{0, 1\}$.

We start by proving an auxiliary lemma that is used both in the proof of Theorem 4.6 and in the proof of Theorem 4.1, presented in the main body, which is the main result of Section 4. It states that the $g$-squared loss of $f$ on $X$ can only be reduced by performing a change of variables to the identity function acting on $f(X)$.

**Lemma A.3.** *For any function $f$, one has $\mathcal{L}_{g,f(X)}(id) \leq \mathcal{L}_{g,X}(f)$.*

PROOF:  Let $\tilde{X} = f(X)$. Note that for a fixed $\tilde{x} \in \tilde{\mathcal{X}}$,

$$\eta_{f^{-1}}(\tilde{x}) = p_{f^{-1}}(1|\tilde{x}) = p\left(1 | X \in f^{-1}(\tilde{x})\right) = \frac{\mathbb{E}_X \eta(X)\mathbf{1}_{\{X \in f^{-1}(\tilde{x})\}}}{\mathbb{E}_X \mathbf{1}_{\{X \in f^{-1}(\tilde{x})\}}}.$$

Hence,

$$\mathcal{L}_{g,f(X)}(id) = \mathbb{E}_{\tilde{x}} \left( (g \circ id)(\tilde{X}) - \eta_{f^{-1}}(\tilde{X}) \right)^2$$

$$= \mathbb{E}_{\tilde{X}} \left( g(\tilde{X}) - \frac{\mathbb{E}_X \eta(X)\mathbf{1}_{\{X \in f^{-1}(\tilde{X})\}}}{\mathbb{E}_X \mathbf{1}_{\{X \in f^{-1}(\tilde{X})\}}} \right)^2$$

$$= \mathbb{E}_{\tilde{X}} \left( \frac{\mathbb{E}_X((g \circ f)(X) - \eta(X))\mathbf{1}_{\{X \in f^{-1}(\tilde{X})\}}}{\mathbb{E}_X \mathbf{1}_{\{X \in f^{-1}(\tilde{X})\}}} \right)^2,$$

since for all $x, x' \in f^{-1}(\tilde{x})$ one has $(g \circ f)(x) = (g \circ f)(x') = g(\tilde{x})$. The Cauchy-Schwarz inequality yields

$$\mathcal{L}_{g,f(X)}(id) \leq \mathbb{E}_{\tilde{X}} \frac{\mathbb{E}_X((g \circ f)(X) - \eta(X))^2\mathbf{1}_{\{X \in f^{-1}(\tilde{X})\}}}{\mathbb{E}_X \mathbf{1}_{\{X \in f^{-1}(\tilde{X})\}}}$$

$$= \mathbb{E}_X((g \circ f)(X) - \eta(X))^2 = \mathcal{L}_{g,X}(f),$$

proving the claim. $\qquad\square$

We conclude this section by proving Theorem 4.6, the final ingredient for connecting the convergence rates of a kNN classifier with the Bayes error in the original space.

PROOF OF THEOREM 4.6:  As in the proof of Lemma 4.3, we know that

$$\Delta^*_{f,X} \leq \mathbb{E}_X \left( p(y_x \mid x) - p_{f^{-1}}(y_x \mid f(x)) \right) \leq \mathbb{E}_X \left| \eta(X) - \eta_{f^{-1}}(f(X)) \right|.$$

The triangle and the Cauchy-Schwarz inequality, once for each term, yield

$$\Delta^*_{f,X} \leq \mathbb{E}_X |\eta(X) - (g \circ f)(X)| + \mathbb{E}_X \left| (g \circ f)(X) - \eta_{f^{-1}}(f(X)) \right|$$

$$= \mathbb{E}_X |\eta(X) - (g \circ f)(X)| + \mathbb{E}_{\tilde{X}} \left| g(\tilde{X}) - \eta_{f^{-1}}(\tilde{X}) \right|$$

$$\leq \left( \underbrace{\mathbb{E}_X |\eta(X) - (g \circ f)(X)|^2}_{\mathcal{L}_{g,X}(f)} \right)^{1/2} + \left( \underbrace{\mathbb{E}_{\tilde{X}} \left| g(\tilde{X}) - \eta_{f^{-1}}(\tilde{X}) \right|^2}_{\mathcal{L}_{g,f(X)}(id)} \right)^{1/2}.$$

The claim now follows by Lemma A.3. $\qquad\square$

## A.2 Convergence Rates of a kNN Classifier over Transformed Features

We now present the proof of Theorem 4.8, mimicking the proof of Theorem 6.2 from [18]. We insert our (weaker) probabilistic Lipschitz assumption where appropriate. It allows us to remove any additional constraint on $f$, leaving us with a statement dependent only on $\mathcal{L}_{g,X}(f)$. As discussed in Section 5, for $g(x) = \mathrm{softmax}(W^T x + b)$ this can be used to rank various transformations $f$ by simply reporting the mean squared error of the test set, denoted by $MSE_g(f, W, b)$. The price we need to pay is an additional additive error term. However, since an unavoidable error term as a function of $\mathcal{L}_{g,X}(f)$ already exists in Theorem 4.6, we accept it here, having in mind the flexibility it gives us. Optimizing this additive error term could form an interesting path for further research.

PROOF OF THEOREM 4.8: It is well known (see Chapter 1 in [18]) that

$$\mathbb{E}_n[(R_X)_{n,k}] - R_X^* \leq 2\mathbb{E}_n\mathbb{E}_X \left| \eta_{n,k}(X) - \eta(X) \right| \leq 2\sqrt{\mathbb{E}_n\mathbb{E}_X \left| \eta_{n,k}(X) - \eta(X) \right|^2}, \quad \text{(A.3)}$$

where the last inequality is a simple application of the Cauchy-Schwarz inequality. With the assumptions as above, it suffices to prove that for all $w \in \mathbb{R}^d$,

$$\mathbb{E}_n\mathbb{E}_X \left| \eta_{n,k}(X) - \eta(X) \right|^2 \leq \frac{1}{k} + cL \left( \frac{k}{n} \right)^{2/d} + \delta + 2\varepsilon^2, \quad \text{(A.4)}$$

for some $c > 0$. Let $(X_1, Y_1), \ldots, (X_n, Y_n)$ be the set of $n$-samples distributed using $p(x, y)$. For $x \in \mathcal{X}$, let $n(i, x)$ denote the index of the $i$-th nearest neighbor of $x$ in $X_1, \ldots, X_n$. Then

$$\mathbb{E}_n \left| \eta_{n,k}(x) - \eta(x) \right|^2 = \underbrace{\mathbb{E}_n \left| \eta_{n,k}(x) - \frac{1}{k}\sum_{i \in [k]} \eta(X_{n(i,x)}) \right|^2}_{J_1(x)} + \underbrace{\mathbb{E}_n \left| \frac{1}{k}\sum_{i \in [k]} \eta(X_{n(i,x)}) - \eta(x) \right|^2}_{J_2(x)}.$$

For $J_1(x)$ note that

$$J_1(x) = \mathbb{E}_n \left| \frac{1}{k}\sum_{i \in [k]} \left( \eta_{n,k}(x) - \eta(X_{n(i,x)}) \right) \right|^2 = \frac{1}{k^2}\sum_{i \in [k]} \mathbb{E}_n \left| Y_{n(i,x)} - \eta(X_{n(i,x)}) \right|^2 \leq \frac{1}{k}. \quad \text{(A.5)}$$

For $J_2(x)$ we have

$$\mathbb{E}_X J_2(X) = \mathbb{E}_X\mathbb{E}_n \left| \frac{1}{k}\sum_{i \in [k]} \left( \eta(X_{n(i,X)}) - \eta(X) \right) \right|^2 \leq \frac{1}{k}\sum_{i \in [k]} \mathbb{E}_n\mathbb{E}_X \left| \eta(X_{n(i,X)}) - \eta(X) \right|^2,$$

by the Cauchy-Schwarz inequality. Let $\mathtt{GOOD}_{\varepsilon,L} := \{(X, X') : |\eta(X) - \eta(X')| \leq \varepsilon + L\|X - X'\|\}$. Since $\eta$ is $(\varepsilon, \delta, L)$-probably Lipschitz and $(a + b)^2 \leq 2a^2 + 2b^2$, we have that

$$\mathbb{E}_X J_2(X) \leq \frac{1}{k}\sum_{i \in [k]} \mathbb{E}_n \left( 1 - \mathbb{P}\left( (X, X_{n(i,X)}) \in \mathtt{GOOD}_{\varepsilon,L} \right) \right) + \frac{1}{k}\sum_{i \in [k]} \mathbb{E}_n\mathbb{E}_X \left( 2\varepsilon^2 + 2L^2\|X_{n(i,X)} - X\|^2 \right)$$

$$\leq \delta + 2\varepsilon^2 + 2L^2\mathbb{E}_X \underbrace{\mathbb{E}_n \frac{1}{k}\sum_{i \in [k]} \left\| X_{n(i,X)} - X \right\|^2}_{J_3(X)}.$$

The term $J_3(X)$ is exactly the same as the upper bound for $I_2(X)$ in the proof of Theorem 6.2 in [18], where it is shown that there exists a $c > 0$ such that $\mathbb{E}_X J_3(X) \leq c(k/n)^{2/d}$.

Combining the bounds for $J_1$, $J_2$ and $J_3$ proves the claim. $\qquad\square$

The final result of this section establishes the probabilistic Lipschitz condition in terms of the $g$-squared error of $f$. It is the glue that brings all the pieces together, having in mind that it is applied on $f(X)$.

PROOF OF LEMMA 4.9: Note that the triangle inequality implies

$$|\eta(X) - \eta(X')| \leq \underbrace{|\eta(X) - g(X)|}_{I_1(X)} + \underbrace{|g(X) - g(X')|}_{I_2} + \underbrace{|g(X') - \eta(X')|}_{I_1(X')}.$$

For $I_2$ note that the fact that $g$ is $L$-Lipschitz implies $I_2(X, X') \leq L\|X - X'\|$.

For $I_1(X), I_1(X')$ we start by defining $\texttt{GOOD}_t := \{x \in \mathcal{X} \colon |\eta(x) - g(x)| \leq t\}$. Note that Markov's inequality yields

$$\mathbb{P}(X \notin \texttt{GOOD}_t) = \mathbb{P}\left(|\eta(X) - g(X)|^2 \geq t^2\right) \leq \frac{\mathcal{L}_{g,X}(id)}{t^2}.$$

Therefore,

$$\mathbb{P}\left(|\eta(X) - \eta(X')| \leq \varepsilon + L\|X - X'\|\right) \geq \mathbb{P}(X, X' \in \texttt{GOOD}_{\varepsilon/2})$$
$$\geq \left(1 - \frac{4\mathcal{L}_{g,X}(id)}{\varepsilon^2}\right)^2 \geq 1 - \frac{8\mathcal{L}_{g,X}(id)}{\varepsilon^2},$$

concluding the proof. □

# B    Extended Experimental Evaluation

As described in the main body of the paper, in this section we report additional experiments and outline the full experimental setup.

## B.1    Experimental Setup

The code to reproduce the results and the graphs from the entire paper is made available in the supplementary material.

**Feature Transformations.** We provide the list of all tested feature transformations, together with their dimensionality, for the vision datasets and text classification datasets in Tables 2 and 3, respectively. We were not able to export the BOW (and hence neither the BOW-TFIDF nor the PCA transformed) feature representations for YELP due to the large amount of samples and their high dimensionality. Additionally, calculating the NCA representations did not successfully terminate for any of the text classification datasets, as this method does not scale to high dimensional and large-sample-size inputs. All reported transformations are publicly available through either the scikit-learn toolkit[6], TensorFlow Hub[7] or PyTorch Hub[8].

**Datasets.** We use the standard splits provided by the datasets, as given in Table 1 in the main body. We collected all the datasets but YELP from the Tensorflow Datasets collection[9], whereas YELP can be downloaded from `https://www.yelp.com/dataset`.

**kNN Classifier.** In order to illustrate the convergence rates, we subsample the training samples 10 times linearly (decreasingly), and perform 30 independent runs in order to report the variance. We plot the 95% confidence intervals on all the convergence graphs.

**Logistic Regression Classifier.** We train all the logistic regression models (on all the datasets and transformations mentioned earlier) using SGD with a momentum value of $0.9$ and a batch size of $64$ on the entire training set for 200 epochs, minimizing the cross entropy loss. We report the best achieved test set error (misclassification error) and mean squared error (MSE) using different values of $L_2$ regularizer $(0.0, 0.0001, 0.001, 0.01, 0.1)$ and initial learning rates $(0.0001, 0.001, 0.01, 0.1)$. We pre-process the input before training by normalizing the features to range between -1 and 1.

**Training infrastructure.** Training of the logistic regression models and evaluating kNN was executed on a single NVIDIA Titan Xp GPU.

Table 2: Feature transformations for images as features.

| Transformation | Source | MNIST | CIFAR10 | CIFAR100 |
|---|---|---|---|---|
| *Identity - Raw* | - | ✓ | ✓ | ✓ |
| PCA (d=32) | scikit-learn | ✓ | ✓ | ✓ |
| PCA (d=64) | scikit-learn | ✓ | ✓ | ✓ |
| PCA (d=128) | scikit-learn | ✓ | ✓ | ✓ |
| NCA (d=64) | scikit-learn | ✓ | ✓ | ✓ |
| AlexNet(d=4096) | PyTorch-Hub | ✓ | ✓ | ✓ |
| GoogleNet (d=1024) | PyTorch-Hub | ✓ | ✓ | ✓ |
| VGG16 (d=4096) | PyTorch-Hub | ✓ | ✓ | ✓ |
| VGG19 (d=4096) | PyTorch-Hub | ✓ | ✓ | ✓ |
| ResNet50-V2 (d=2048) | TF-Hub | ✓ | ✓ | ✓ |
| ResNet101-V2 (d=2048) | TF-Hub | ✓ | ✓ | ✓ |
| ResNet152-V2 (d=2048) | TF-Hub | ✓ | ✓ | ✓ |
| InceptionV3 (d=2048) | TF-Hub | ✓ | ✓ | ✓ |
| EfficientNet-B0 (d=1280) | TF-Hub | ✓ | ✓ | ✓ |
| EfficientNet-B1 (d=1280) | TF-Hub | ✓ | ✓ | ✓ |
| EfficientNet-B2 (d=1408) | TF-Hub | ✓ | ✓ | ✓ |
| EfficientNet-B3 (d=1536) | TF-Hub | ✓ | ✓ | ✓ |
| EfficientNet-B4 (d=1792) | TF-Hub | ✓ | ✓ | ✓ |
| EfficientNet-B5 (d=2048) | TF-Hub | ✓ | ✓ | ✓ |
| EfficientNet-B6 (d=2304) | TF-Hub | ✓ | ✓ | ✓ |
| EfficientNet-B7 (d=2560) | TF-Hub | ✓ | ✓ | ✓ |

Table 3: Feature transformations for natural language as features.

| Transformation | Source | IMDB | SST2 | YELP |
|---|---|---|---|---|
| *Identiy - BOW* | - | ✓ | ✓ | ✗ |
| BOW-TFIDF | scikit-learn | ✓ | ✓ | ✗ |
| PCA (d=8) | scikit-learn | ✓ | ✓ | ✗ |
| PCA (d=16) | scikit-learn | ✓ | ✓ | ✗ |
| PCA (d=32) | scikit-learn | ✓ | ✓ | ✗ |
| PCA (d=64) | scikit-learn | ✓ | ✓ | ✗ |
| PCA (d=128) | scikit-learn | ✓ | ✓ | ✗ |
| ELMO (d=1024) | TF-Hub | ✓ | ✓ | ✓ |
| NNLM-EN (d=50) | TF-Hub | ✓ | ✓ | ✓ |
| NNLM-EN-WITH-NORMALIZATION (d=50) | TF-Hub | ✓ | ✓ | ✓ |
| NNLM-EN (d=128) | TF-Hub | ✓ | ✓ | ✓ |
| NNLM-EN–WITH-NORMALIZATION (d=128) | TF-Hub | ✓ | ✓ | ✓ |
| Universal Sentence Encoder (USE) (d=512) | TF-Hub | ✓ | ✓ | ✓ |
| BERT-Base (d=678) | PyTorch-Hub | ✓ | ✓ | ✓ |

## B.2 Convergence Plots

We provide convergence plots for an interesting subset of the datasets (CIFAR100 and IMDB) and transformations in Figures 5 and 6. From the results and scripts that we made available through the supplementary materials, one could simply create and analyze the plots for arbitrary combination of considered datasets and transformations. We remark that on both plots the transformations that achieve the best possible convergence in the finite sample regime do not have the lowest dimension. Furthermore, the starting point of the convergence lines for such transformations is typically much lower than the starting point of standard dimension-reduction techniques such as PCA. Having access to much more (ideally infinitely many) training samples would result in every line converging to the final, irreducible-bias term per transformation.

Figure 5: Impact of the dimension on CIFAR100 using all involved transformations **(Left)**, and PCA-based transformation only **(Right)**.

Figure 6: Impact of the dimension on IMDB using all involved transformations **(Left)**, and PCA-based transformation only **(Right)**.

### B.3 On the Impact of the Hyper-Parameter k

It is well known that one can choose the hyper-parameter $k$ to reach the best possible convergence in the finite data regime depending on the dataset. We investigate this with respect to transformations by showing that different transformations on the same dataset might have different optimal choices for $k$. This tradeoff for a fixed dataset is not clearly visible in the main Theorem 4.1 due to the usage of $\mathcal{O}(\cdot)$ notation, hiding the constants. However, by exploring the proof outline and analyzing (A.5), one realizes that the upper bound of $J_1$ is dependent on the posterior in the transformed feature space, which might change for a fixed input dataset. We report the empirically observed minimal kNN test error for values of $k$ ranging from 1 to 250 in Table 4, for all the feature transformations on the computer vision datasets, and in Table 5, for the all the text classification datasets. In practice, when using kNN, one would take a portion of the training set as a validation set to choose the best hyper-parameter value for $k$ and run an evaluation of the test set in order to control overfitting.

Table 4: Minimal kNN errors for the computer vision datasets.

| Transformation | MNIST | CIFAR10 | CIFAR100 |
|---|---|---|---|
| *Identiy - Raw* | 0.029 (k=3) | 0.646 (k=1) | 0.825 (k=1) |
| PCA (d=32) | **0.025 (k=8)** | 0.575 (k=16) | 0.811 (k=1) |
| PCA (d=64) | **0.025 (k=3)** | 0.601 (k=18) | 0.806 (k=1) |
| PCA (d=128) | 0.028 (k=3) | 0.619 (k=1) | 0.810 (k=1) |
| NCA (d=64) | 0.026 (k=5) | 0.600 (k=18) | 0.837 (k=39) |
| AlexNet | 0.165 (k=13) | 0.244 (k=13) | 0.509 (k=19) |
| GoogleNet | 0.113 (k=9) | 0.171 (k=10) | 0.431 (k=18) |
| VGG16 | 0.133 (k=16) | 0.208 (k=19) | 0.476 (k=15) |
| VGG19 | 0.138 (k=15) | 0.205 (k=19) | 0.470 (k=16) |
| ResNet50-V2 | 0.092 (k=5) | 0.152 (k=9) | 0.397 (k=17) |
| ResNet101-V2 | 0.092 (k=6) | 0.126 (k=9) | 0.371 (k=10) |
| ResNet152-V2 | 0.094 (k=3) | 0.137 (k=6) | 0.373 (k=14) |
| InceptionV3 | 0.049 (k=13) | 0.150 (k=10) | 0.407 (k=17) |
| EfficientNet-B0 | 0.535 (k=7) | 0.159 (k=9) | 0.410 (k=17) |
| EfficientNet-B1 | 0.691 (k=7) | 0.125 (k=7) | 0.368 (k=24) |
| EfficientNet-B2 | 0.630 (k=8) | 0.120 (k=10) | 0.352 (k=10) |
| EfficientNet-B3 | 0.789 (k=7) | 0.090 (k=6) | 0.312 (k=13) |
| EfficientNet-B4 | 0.745 (k=25) | **0.085 (k=7)** | **0.307 (k=13)** |
| EfficientNet-B5 | 0.804 (k=23) | 0.092 (k=8) | 0.317 (k=12) |
| EfficientNet-B6 | 0.649 (k=25) | 0.092 (k=6) | 0.326 (k=10) |
| EfficientNet-B7 | 0.543 (k=14) | 0.087 (k=12) | 0.316 (k=9) |

Table 5: Minimal kNN errors for the text classification datasets.

| Transformation | IMDB | SST2 | YELP |
|---|---|---|---|
| *Identiy - BOW* | 0.334 (k=36) | 0.349 (k=6) | - |
| BOW-TFIDF | 0.243 (k=247) | 0.249 (k=26) | - |
| PCA (d=8) | 0.274 (k=155) | 0.408 (k=81) | - |
| PCA (d=16) | 0.226 (k=159) | 0.382 (k=236) | - |
| PCA (d=32) | 0.216 (k=175) | 0.375 (k=174) | - |
| PCA (d=64) | 0.228 (k=157) | 0.377 (k=147) | - |
| PCA (d=128) | 0.241 (k=196) | 0.374 (k=170) | - |
| ELMO | 0.255 (k=37) | **0.195 (k=206)** | 0.424 (k=166) |
| NNLM (d=50) | 0.287 (k=77) | 0.294 (k=227) | 0.475 (k=86) |
| NNLM (d=128) | 0.255 (k=57) | 0.241 (k=148) | 0.452 (k=43) |
| NNLM (d=50, w/ normalize) | 0.259 (k=45) | 0.288 (k=36) | 0.455 (k=92) |
| NNLM (d=128, w/ normalize) | 0.227 (k=47) | 0.241 (k=162) | 0.427 (k=45) |
| Universal Sentence Encoder (USE) | **0.188 (k=183)** | 0.201 (k=186) | **0.387 (k=75)** |
| BERT-Base | 0.266 (k=45) | 0.267 (k=8) | 0.441 (k=51) |

## Footnotes

[6] `https://scikit-learn.org/stable/`

[7] `https://tfhub.dev/`

[8] `https://pytorch.org/hub/`

[9] `https://www.tensorflow.org/datasets/`


[Supplementary Material 2 · all_vision_transfer.pdf]



# MNIST

R: 0.162

# CIFAR10

R: -0.396

# CIFAR100

R: -0.366

1NN Error vs Dimension (top row) and 1NN Error vs MSE (bottom row)

MNIST: R (w/o Norm): 0.962, R (CCA): 0.964

CIFAR10: R (w/o Norm): 0.992, R (CCA): 0.992

CIFAR100: R (w/o Norm): 0.993, R (CCA): 0.993

Norm (MNIST): 10.30, 20.62, 39.17, 58.28

Norm (CIFAR10): 4.72, 15.66, 20.25, 24.27

Norm (CIFAR100): 13.42, 39.52, 48.52, 100.33

Legend: EfficientNet-B0, EfficientNet-B1, EfficientNet-B2, EfficientNet-B3, EfficientNet-B4, EfficientNet-B5, EfficientNet-B6, EfficientNet-B7, GoogleNet, InceptionV3, NCA (d=64), PCA (d=128), PCA (d=32), PCA (d=64), RAW, ResNet-101, ResNet-152, ResNet-50, VGG16, VGG19

[Supplementary Material 3]



**IMDB** | **SST2** | **YELP**

Top row:
- 1NN Error vs Dimension
- IMDB: R: -0.362
- SST2: R: -0.628
- YELP: R: -0.485

Bottom row:
- 1NN Error vs MSE
- IMDB: R (w/o Norm): 0.419, R (CCA): 0.541
- SST2: R (w/o Norm): 0.869, R (CCA): 0.917
- YELP: R (w/o Norm): 0.843, R (CCA): 0.936

Norm (IMDB): 2.72, 7.56, 10.15, 11.01
Norm (SST2): 1.30, 3.84, 6.82, 10.97
Norm (YELP): 6.39, 20.59, 26.40, 27.68

Legend:
- ELMO
- NNLM (d=128)
- NNLM (d=128, w/ normalize)
- NNLM (d=50)
- NNLM (d=50, w/ normalize)
- PCA (d=128)
- PCA (d=16)
- PCA (d=32)
- PCA (d=64)
- PCA (d=8)
- USE

[Supplementary Material 4 · cifar100_convergence.pdf]



**CIFAR100 - All**

**CIFAR100 - RAW & PCA**

Y-axis (left): 1NN Error — 0.4, 0.6, 0.8
X-axis (left): Training Samples — 20000, 40000

Y-axis (right): 0.78, 0.80, 0.82, 0.84, 0.86
X-axis (right): Training Samples — 30000, 35000, 40000, 45000, 50000

Legend:
- - - RAW (d=3072)
- - - GoogleNet (d=1024)
- - - InceptionV3 (d=2048)
- - - EfficientNet-B7 (d=2560)
- - - NCA (d=64)
- - - PCA (d=32)
- - - PCA (d=64)
- - - PCA (d=128)

[Supplementary Material 5]



**IMDB - All**

**IMDB - BOW & PCA**

1NN Error (y-axis, left panel): 0.30, 0.35, 0.40

Training Samples (x-axis, left panel): 10000, 20000

1NN Error (y-axis, right panel): 0.325, 0.350, 0.375, 0.400

Training Samples (x-axis, right panel): 15000, 17500, 20000, 22500, 25000

Legend:
- BOW (d=104K)
- BOW-TFIDF (d=104K)
- BERT (d=768)
- ELMO (d=1024)
- USE (d=512)
- PCA (d=8)
- PCA (d=16)
- PCA (d=32)