[Reviews · NeurIPS 2020]

Review 1

Summary and Contributions: Update: Thanks for addressing the concerns raised by the reviewers, based on re-reading the paper and going over the comments, I am able to understand the experiments better - and based on the authors comments that they will revise the draft to make things more clear, I will change my score to accept. Having said that, I would still keep my confidence low since I am unable to accurately access the significance of the result and I believe that would be a key factor to consider in a novel theoretical paper. ------------------------------------------------- The paper provides a theoretical result of an upper bound on the convergence rate of kNN classifiers over feature transformations. The result is based on two key properties of the transformed space that they identify. The first is 'safety', which is a measure of how well can we recover the posterior in the original space from the feature space. The second is smoothness, which is a measure of how hard it is to recover the posterior in the original space from the feature space.

Strengths: The paper seems to address a novel issue of bridging the gap between theoretical understanding of kNN and practical application. They provide a theoretical upper bound on the convergence.

Weaknesses: While there are experiments presented, the results could have been presented with more clarity. I couldn't see the significance of the result immediately in the experiments presented, perhaps a more elaborate explanation would be helpful.

Correctness: They appear to be correct but I have not verified the proofs step by step.

Clarity: I had some issues with certain parts of the paper - it would be helpful if the authors provided definitions of all notations used.

Relation to Prior Work: They have mentioned previous work on kNN but they state that the paper covers a novel analysis and is one of the first papers in this direction.

Reproducibility: Yes

Additional Feedback: Could you please elaborate on how the definitions of safety and smoothness relate to the representations as loss the Lipschitz constant respectively? In section 5, you have mentioned that there is significant improvement in correlation on all the test classification datasets. Could you elaborate on this further?


Review 2

Summary and Contributions: This paper analyzes k-NN classification error when the feature space has been transformed from its original raw space to, for instance, a learned embedding space. Importantly the resulting bounds depend on a notion of "safety" (how well the raw feature space's posterior probability function can be recovered) and smoothness (Lipschitz constant related to the recovery function).

Strengths: This work addresses a major gap between theory and practice of k-NN methods: nowadays, k-NN classification is typically applied in a learned embedding space (e.g., a penultimate or bottleneck layer of a deep net) rather than in the raw feature space, but theory for k-NN methods is typically stated for the raw feature space or alternatively only in the transformed space (if one treats the transformation as a pre-processing step, so that the transformation itself doesn't appear in the theory except implicitly in the metric). This paper is the first one that I'm aware of that explicitly characterizes error, accounting for there being raw and transformed feature spaces as well as a transformation function. This is a significant advance in theory for nonparametric methods. The authors do a good job motivating and explaining the theory especially in illustrative examples of how transformations affect the Bayes error rate and how this plays into the definition of delta-safety.

Weaknesses: It would be very helpful getting a sense of how tight the theoretical bounds are, if not in theory then in numerical simulations (for example in a toy dataset where the true posterior probability function is known) with different choices of transformations. I think the experimental results can be improved by explicitly also looking at different values of k (without optimizing over k, which is what seems to be done in the supplemental material), especially since your bound has the O(k^{-1/2}) and O(||w|| (k/n)^{1/d}) terms as well. Then you can use CCA to find a linear combination of k^{-1/2}, ||w|| (k/n)^{1/d}, and MSE^{1/4} to try to explain the k-NN error (if I understand how you're using CCA, basically by choosing k=1, you can just look at ||w|| and MSE^{1/4}?). It seems like picking k=1 perhaps doesn't yield a good bound (unless it turns out that in practice that first O(k^{-1/2}) term isn't that important). It would be helpful to understand how much the k^{-1/2} and (k/n)^{1/d} terms matter.

Correctness: The claims and experimental setup look correct.

Clarity: The paper is overall well-written. The paragraph in lines 294-297 could be written a bit more clearly (also there's a typo there where "later" should be "latter"); from my understanding the linear combination of the MSE and norm is determined via CCA, although it seems like you might actually want MSE^{1/4}?

Relation to Prior Work: Yes, the authors place their work in the context of existing literature.

Reproducibility: Yes

Additional Feedback: Update (after author feedback/reading other reviews): Overall, it seems like the experimental results could use polishing in presentation/experiments. The author feedback makes it clear that they're aware of the issues and have fixes in mind. I strongly encourage the authors to revise the experiments section to make things more clear, as they indicated in their response. I am leaving my score the same, and still think that this paper should be accepted.


Review 3

Summary and Contributions: This paper proposes a novel convergence analysis of kNN over transformed features. This work is novel and fills the gap between the applications and theoretical analysis where most are derived for the raw feature space. The authors then propose to evaluate the convergence bound based on the safety and smoothness of the transformed space. Then empirical evaluations are conducted with 30 feature transformations on image and text data.

Strengths: + The proposed analysis is novel. While most previous analysis focuses on the raw feature space, most applications use feature transformation such as PCA and pre-trained neural network. This mismatch leads to a gap between the theoretical analysis and applications where this work tries to tackle. + The theoretical analysis is sound. The authors provide detailed proof and analysis. + The writing is clear and easy to follow.

Weaknesses: - The experimental results are not significant. Two straightforward error rate indicators: the dimension of the space and LR error is chosen as baselines and the authors claim that the proposed bound is superior to these two baselines. However, from the experimental results, Pearson’s r correlation in Fig.4 shows that the proposed improvement is marginal. The results on image dataset from the supplementary material also shows that the Pearson’s r correlation is very close between the LR Error and MSE. - The authors only conduct experiments with k=1. It is unclear whether the empirical conclusion remains the same in Fig.4 if k becomes larger than 1. - The figures are not clear enough, e.g. what does each point stand for in Fig.1 and Fig.2. The authors could add the necessary information to make this paper more self-contained.

Correctness: The claims and methods are correct. The empirical methodology is correct but not convincing enough.

Clarity: Yes, this paper is clearly written.

Relation to Prior Work: yes

Reproducibility: Yes

Additional Feedback: The authors could provide further analysis of the close performances between LR Error and MSE. ****************updated review**************** I was convinced by authors' rebuttal and would like to raise my score to "above the acceptance threshold".

[Author Response · NeurIPS 2020]

We thank the reviewers for their insightful comments which we will integrate in the final version of the paper.

We appreciate reviewers' consensus that this work provides a novel theoretical study. Given this novelty, we are also
well aware that this work opens many questions and answering all of them requires future work. Being the first work
that aims at understanding the influence of transformations on kNN, we see this work as a starting point of a very
interesting research area.

**Common Concern (Reviewer 1, Reviewer 4): Significance of Empirical Result.** One common, major concern from
R1 and R4 is the significance of the empirical result. We agree that the current way the empirical results are presented
is confusing, and hope to clarify and address it here.

The contribution of this work is to establish the following result: Both the MSE error *and* the smoothness are important
factors governing the convergence of KNN over transformations; to the surprise of a common belief that the dimension
is important. As a result, to verify the significance of our contribution, we need to establish that:

1. Using MSE leads to significantly better correlation compared with using the dimension;
2. Examining the smoothness (represented by the norm), in addition to MSE, can lead to further improvement.

In terms of the comparison with LR Error and the fact that MSE seems to give only slightly better result than LR
Error – we agree that this is confusing and we apologize for it. LR Error is just a practical metric that is similar to
the MSE, and is included since LR Error is popular alternative of MSE, often used in practice (therefore, the similar
performance between the MSE error and the LR error is expected). However, **there is no theoretical understanding**
**linking LR Error to KNN convergence over feature transformations**. As a result, LR Error is not the baseline that
we are comparing with. The significance of our contribution is between (1) MSE + Norm, (2) MSE, and (3) Dimension.
We want to express again our appreciation to the reviewers for this very constructive feedback and we will revise our
draft accordingly to reflect this.

**Common Concern (Reviewer 2, Reviewer 4): Empirical Results for $k > 1$.** We agree that understanding the behav-
ior of KNN for different $k$ is important. We performed experiments for $k > 1$, over all datasets and all embeddings,
and the result were uploaded with the initial submission as supplementary material[1]. **We want to emphasize that all**
**the results for $k > 1$ confirm the theoretical findings, whilst for some $k$ they offer an improvement in the linear**
**correlation** (for example, SST2 one can get CCA score up to 0.97 when $k = 8$, whereas in the main body we report
0.917, for $k = 1$). When writing the submitted version of the paper we opted for $k = 1$ since it already serves the
purpose of showing that MSE is important on its own, whilst one could see that including the smoothness further
improves the correlation. However, we are now aware that we should summarize the findings for $k > 1$ in the main
body of the paper. We will do for the final version of the paper and we thank the reviewers for pointing this out.

We further address each reviewer individually.

**Reviewer 1.** Intuitively, *safety* explains how much of information is preserved after applying a transformation, with
Theorem 4.6 showing that it can be controlled by the $L^2$ loss. *Smoothness* is a common assumption in the work on
convergence rates of nearest neighbor estimators which allows the new point to learn from its neighbors. It is usually
given through the Lipschitz constant and we explore this in Definition 4.7 and Theorem 4.8. We paid particular attention
to defining all the necessary notions, in particular to novel definitions, and we will do a final check for the final version.

**Reviewer 2.** We agree with the reviewer that one could indeed examine the tightness of our theoretical bounds by
constructing a toy dataset with a known true posterior probability and a set of transformations. However, such a
transformation should either be an identity (if the toy dataset already has the desired property), or carefully constructed
for this purpose only. In this paper we opted for not constructing a single transformation ourselves, as our main goal is
to bridge the gap between the real-world applications (e.g. public pre-trained embeddings) with the theory of nearest
neighbors. For example, even Lemma 4.4, which establishes a sharp bound for the safety, works for any transformation.
Our probabilistic Lipschitz condition is as weak as possible and we believe that establishing tight bounds will form
an interesting future work. To this end, we believe that stronger assumptions would yield a better exponent in $MSE$,
which is why for simplicity we opted for presenting $MSE$ instead of $MSE^{1/4}$ in the paper. We thank the reviewer for
suggesting the study of CCA of $k^{-1/2}$, $\|w\|(k/n)^{1/d}$ and $MSE^{1/4}$ (or some other exponent), since it could be another
way of understanding the tightness of the bound for $k > 1$, providing an interesting future direction. **With all that in**
**mind, we will definitely include the above reasoning on the tightness and the challenges involved in evaluating**
**it, as a discussion in the final version of this manuscript.**

**Reviewer 4.** We agree that the figures should be more self-contained and we will address this in the final version.

## Footnotes

[1] file code/results/*<DATASET>*/knn_accuracy/*<EMBEDDING>*.csv


[Meta-Review · NeurIPS 2020]

This paper provides some interesting theoretical insights into the convergence of kNN over feature transformations. This is backed up by some empirical results. All three reviewers argue for acceptance, but have also provided some directions for improvement, which was acknowledged by the authors in their feedback, promising to include these changes in the final version. Personally I have one issue with the paper, which is introducing some datasets in the experimental section, without providing any results. These are supplied in the additional material, to me that feels like cheating. On the other hand, the conference allows supplements, so kind of encourages it.